# Transfer of knowledge from model organisms to evolutionarily distant non-model organisms: The coral *Pocillopora damicornis* membrane signaling receptome

Lokender Kumar[1], Nathanael Brenner[1], Samuel Sledzieski[2], Monsurat Olaosebikan[3], Liza M. Roger[4], Matthew Lynn-Goin[1], Roshan Klein-Seetharaman[5], Bonnie Berger[2], Hollie Putnam[6], Jinkyu Yang[7], Nastassja A. Lewinski[4], Rohit Singh[2], Noah M. Daniels[8], Lenore Cowen[3], Judith Klein-Seetharaman[1]*

**1** Department of Chemistry, Colorado School of Mines, Golden, CO, United States of America, **2** MIT Computer Science & Artificial Intelligence Lab, Massachusetts Institute of Technology, Cambridge, MA, United States of America, **3** Department of Computer Science, Tufts University, Medford, MA, United States of America, **4** Department of Chemical and Life Science Engineering, Virginia Commonwealth University, Richmond, VA, United States of America, **5** Yale University, New Haven, CT, United States of America, **6** Department of Biological Sciences, University of Rhode Island, South Kingstown, RI, United States of America, **7** Department of Department of Aeronautics & Astronautics, University of Washington, Seattle, WA, United States of America, **8** Department of Computer Science and Statistics, University of Rhode Island, South Kingstown, RI, United States of America

* jkleinse@asu.edu

## Abstract

With the ease of gene sequencing and the technology available to study and manipulate non-model organisms, the extension of the methodological toolbox required to translate our understanding of model organisms to non-model organisms has become an urgent problem. For example, mining of large coral and their symbiont sequence data is a challenge, but also provides an opportunity for understanding functionality and evolution of these and other non-model organisms. Much more information than for any other eukaryotic species is available for humans, especially related to signal transduction and diseases. However, the coral cnidarian host and human have diverged over 700 million years ago and homologies between proteins in the two species are therefore often in the gray zone, or at least often undetectable with traditional BLAST searches. We introduce a two-stage approach to identifying putative coral homologues of human proteins. First, through remote homology detection using Hidden Markov Models, we identify candidate human homologues in the cnidarian genome. However, for many proteins, the human genome alone contains multiple family members with similar or even more divergence in sequence. In the second stage, therefore, we filter the remote homology results based on the functional and structural plausibility of each coral candidate, shortlisting the coral proteins likely to have conserved some of the functions of the human proteins. We demonstrate our approach with a pipeline for mapping membrane receptors in humans to membrane receptors in corals, with specific focus on the stony coral, *P. damicornis*. More than 1000 human membrane receptors mapped to 335 coral receptors, including 151 G protein coupled receptors (GPCRs). To

**Data Availability Statement:** All relevant data are within the paper and its Supporting Information files.

**Funding:** This work was supported by NIH grant number R35GM141861, awarded to R.S., S.S., and B.B., the National Science Foundation Office of Advanced Cyberinfrastructure award numbers 1940169 to J.K.S., 1939263 to L.C., 1939699 to N. L., 1939795 to H.P., 1939249 to J.K.Y., National Science Foundation Division of Computing and Communication Foundations award number 2029543 to J.K.S., National Science Foundation Graduate Research Fellowship 1745302 to SS and the National Institute of Food and Agriculture award number 1017848 to H.P. The funders had no role in study design, data collection and analysis, decision to publish, or preparation of the manuscript.

**Competing interests:** The authors have declared that no competing interests exist.

validate specific sub-families, we chose opsin proteins, representative GPCRs that confer light sensitivity, and Toll-like receptors, representative non-GPCRs, which function in the immune response, and their ability to communicate with microorganisms. Through detailed structure-function analysis of their ligand-binding pockets and downstream signaling cascades, we selected those candidate remote homologues likely to carry out related functions in the corals. This pipeline may prove generally useful for other non-model organisms, such as to support the growing field of synthetic biology.

## Introduction

A bioinformatics functional genomics/proteomics pipeline for a newly sequenced non-model eukaryotic organism can (and should) seek to leverage the wealth of known information and annotation available for model species that are evolutionarily related. Due to the urgency of tackling species loss through environmental damage, many new non-model organisms are currently being sequenced, especially corals. Once the likely constituent genes have been identified [1,2], predicting if the relevant evolutionary functions of the non-model species genes are conserved or have diverged from their orthologous counterparts in the model species becomes directly relevant. Of course, there may be genes in the non-model organism that have no homologues, also referred to as the dark genome [3]. However, the goal of this paper is to maximally exploit existing knowledge from model organisms for the mining of non-model organisms for function.This is because this step is often quite difficult for organisms such as corals, where the host is 700 million years distant from the closest well-annotated model organism, even for many of the genes in closely related species to the model organism [4]. It is particularly interesting that coral animals contain a surprising number of human homologs missing from fly and *C. elegans* [5].

Corals are complex organisms consisting of an animal host (cnidarian), endosymbiotic dinoflagellate algae and a microbiome with more than 20,000 species of bacteria, bacteriophages, fungi and viruses, collectively referred to as holobiont [6]. We consider here, as a case study, the membrane receptor proteins in corals (host cnidarian). These are important families of proteins to study in coral cnidarian hosts because they have large potential to mediate the host's interactions with its symbiont, microbiome and environment. Understanding these mechanisms in corals has become urgent. As a result of anthropogenic activities, both local and global, coral reefs (i.e. coral holobiont assemblages) are declining rapidly. Mass coral bleaching, or the expulsion of the symbiotic algae due primarily to thermal and photophysiological stress driven by marine heatwaves and high irradiance, is resulting in substantial coral mortality [7]. The IPCC Sixth Assessment Report states that under the current climate agreements (1.5˚C warming, Paris Agreement signed in 2015) 70–90% of corals worldwide will be lost by 2050, and as much as 99% if warming reaches 2˚C [8]. Thus understanding the relevant parts of the coral animal genome that are relevant to interactions with their environment has become an urgent bioinformatics task, made possible by the recent availability of coral genomes for multiple species of both animal and symbiont. Fortunately, large classes of membrane receptor proteins have been preserved over time [9], so there is a large body of prior knowledge that we can extrapolate from.

Given a protein of interest that has not been studied before, how much can be learnt from better studied proteins is a common question in biology. The answer lies in the sequence-structure-function paradigm, namely that a similar sequence usually maps to a similar

structure and conserved function. This is the premise for the use of sequence alignments, for which many methods of analysis exist, the choice of which typically depends on how similar the sequences to be compared are.

For many years, BLAST [10] was the *de facto* standard homology search tool, as it provided comparable sensitivity to exact alignment methods such as Smith-Waterman [11] and early heuristics such as FASTA [12] with much faster runtime performance. However, these methods, and even iterative methods such as PSI-BLAST, are unable to detect homologs in the "twilight zone" of homology, between 10 and 30% sequence identity [13,14]. The development of profile Hidden Markov Models (HMMs) such as HMMER [15] and SAM [16] improved remote homology detection performance, but as they still attempt to score a single query sequence against a profile-based HMM, they may miss homologs deep into the "twilight zone." A more recent advancement came in the form of HMM-HMM alignment [17] with HHpred in 2005, with further improvements resulting in HHblits [18]. These methods build a library of HMMs (or use an existing library) from families of nucleotide or protein sequences, and next build a *query* HMM from a multiple sequence alignment based on a BLAST-like search for similar sequences to the query, and use a variant of the Viterbi algorithm [19] to align the two HMMs according to maximum likelihood. HHblits was our choice for remote homology detection in this pipeline, as it combines state-of-the-art sensitivity to remote homologs with fast runtime performance, and is able to detect homologs in the range of 10–30% sequence identity. Given the roughly 700 million years of evolutionary divergence between cnidarians and humans, we could not assume that homologs would be of higher sequence identity. In order to avoid confusion with other organisms, we built a custom HHblits database for *Pocillopora damicornis*, and one of its symbiotic algae, *Cladocopium goreaui (C1)*, formerly known as *Symbiodinium (S.) goreaui* (Clade C, type C1) [20] which we were able to query with human sequences of known function.

We know that corals adjust their behavior in response to external and internal cues, as illustrated by the following examples. Corals contract or extend their tentacles in response to light intensity [21]. Coral larvae are known to prefer red over white surfaces for settling [22]. Corals can fight, so they must be able to sense and attack the enemy or competitor [23]. Corals can distinguish organisms to tolerate (symbiosis) versus subjecting them to an immune attack to prevent disease [24]. Corals prefer to eat plastic over copepods which may relate to a sense of taste [25]. Since corals manage 90% of their energy from symbiotic algae [25,26], they must also be able to measure and regulate nutrient balance. The key question that arises from these observations is: what are the molecular mechanisms underlying these behaviors?

Generally, signal sensing and response reactions in biological systems depend on membrane receptor signaling systems. Receptor activation involves the detection of the signaling molecule (ligand) outside the cell when the ligand binds to the receptor protein present on the surface of the cell (**Fig 1**). The signal transduction stage involves the activation of the receptor (conformational change) leading to the chain reaction of the activation of intracellular proteins. These signal transduction cascades trigger specific cellular responses. Thus, membrane receptors most generally are proteins that are coupled to intracellular signal transduction cascades. There are two types of membrane receptors, sometimes referred to as type I and type II receptors (**Fig 1**). Type I membrane receptors usually contain one or two transmembrane (TM) helices, and often carry enzymatic activity in their cytoplasmic domains, such as tyrosine kinase receptors like the epidermal growth factor receptor. Type II membrane receptors are G-protein coupled receptors (GPCR) which exclusively contain a seven-TM helical bundle. There are also ion channels which change their permeability in response to external signals, but are usually not classified as receptors [27]. Membrane receptors have three major structural domains: extracellular (EC), transmembrane (TM), and cytoplasmic (CP). We will

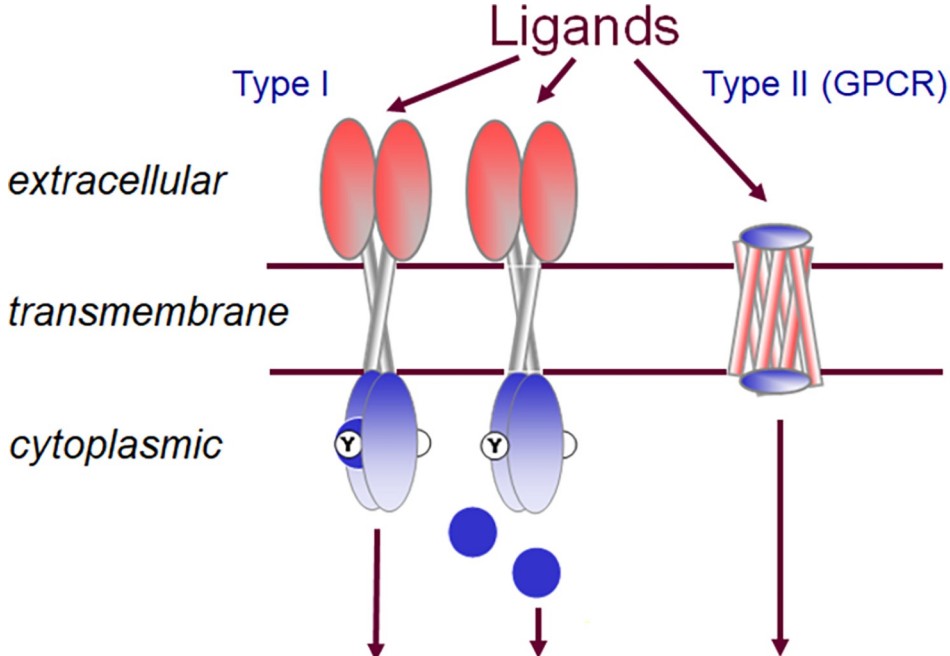

**Fig 1. Schematic representation of type I and type II membrane receptors in signal transduction.** Type I membrane receptors (left) typically possess one transmembrane helix per chain and usually form dimers. They may contain enzymatic activity in the cytoplasmic domain, such as a kinase, or they may recruit a soluble enzyme (blue circle) through protein-protein interaction. Type II membrane receptors are G protein coupled receptors (GPCR) which are characterized by seven transmembrane helices. All receptors (type I and type II) bind to ligands in the extracellular domain, transmit the signal via the transmembrane domain to the cytoplasmic side where they initiate appropriate signal transduction cascades.

discuss one example in depth for each of the two types of membrane receptor proteins. For the type II membrane receptor family, we have chosen the GPCR sub-family of opsins. As an example, for type I membrane receptors, we will discuss Toll-like receptors (TLRs).

GPCRs respond to a large diversity of signals, from light in the opsin family to binding of hormones, neurotransmitters and even other proteins [28]. Correspondingly, in humans, these receptors take a premier role as pharmaceutical targets and have thus been extensively studied [29]. Signal transduction by GPCRs is via binding and activation of heterotrimeric G proteins composed of alpha, beta and gamma subunits [30]. GPCRs are responsible for the majority of cellular responses to external stimuli. Upon activation by a ligand, the receptor promotes the exchange of GTP for GDP in its partner heterotrimeric G protein complex, leading to the dissociation of the alpha and beta gamma complexes from the receptor and each other. The G proteins then interact with other downstream effector proteins to mediate a cell response. In this study, we have used our remote homology detection pipeline to find human GPCR as well as G protein homologs in *P. damicornis* and *C. goreaui*.

TLRs play a crucial role in innate immunity [31]. These membrane glycoprotein receptors recognize and respond to a variety of microbial (viral, fungal, and bacterial) components such as lipopeptides [32], peptidoglycans and lipopolysaccharides [33], flagellin [33,34], DNA [35], and RNA [36]. TLRs consist of leucine-rich repeats (LRR), and the Toll/interleukin-1 receptor (TIR) domain [37]. TLRs initiate signal transduction through interactions with TIR-domain

containing adapters such as MyD88 [38], that in turn recruits interleukin-1 receptor-associated kinase (IRAK) via interactions through death domains [39]. After phosphorylation, IRAK family proteins interact with the TRAF6 adapter. TRAF6 activates TAK1, a member of the mitogen-activated protein (MAP) kinase family [40], leading to the activation of NF-kB via kinase dependent signaling cascade involving IkB kinase complex (IKK-,-β, and -γ) and MAP kinases (ERK, p38, JNK), resulting in the expression of target genes [41]. Viruses and bacteria are abundant in seawater and live in close association with corals [42]. The chemical crosstalk between corals and microbes plays an important role in coral growth and development. Corals need TLRs to communicate with microbes and it has been proposed that TLR signaling is conserved in corals [43]. In the present study, we used our remote homology detection pipeline to find human TLR protein homologs in *P. damicornis* and *C. goreaui* to advance our understanding of coral innate immunity.

Using these two representative membrane receptor families, we will demonstrate that the stated problem of identifying remote homologues in non-model organisms is not just a remote sequence detection problem, but highlights the need for functional investigation of the putative proteins, which involves both systems and structural aspects of these proteins and their interaction partners. To this end, we focus on those positions in the alignments between model and non-model organism homologues that determine specificity of the respective protein's functions. We will discuss below first the common elements of the pipeline, how we determine the pool of all membrane receptors in *P. damicornis*, how we subdivide this broad group into type I and type II membrane receptors, and how we refined these two multi-subfamily groups into functional classes by analysis of two of them, TLRs and opsins, by way of identifying the specificity determination positions, followed by analysis of the downstream signaling proteins.

## Materials and methods

A github repository is available for this project accessible at https://github.com/judithks/corals

### (a) Protein sequence retrieval

All non-*P. damicornis* and *C. goreaui* sequences were retrieved from the UniProt—Swiss-Prot Protein Knowledgebase, SIB Swiss Institute of Bioinformatics; Geneva, Switzerland (https://www.uniprot.org/). *P. damicornis* sequences [44] were downloaded from http://pdam.reefgenomics.org/download/. *C. goreaui* sequences [45] were obtained from http://symbs.reefgenomics.org/download/. P. damicornis sequences are referenced by their numbers only, so for example, pdam_00017423-RA will be referred to as 17423.

Specific subgroups of sequences were identified and retrieved as follows and subjected to HHblits analysis as described in section (b).

**(i) Human membrane receptor list.** We have utilized two human membrane protein lists. The first one was published as the human membrane receptome in 2003 [46]. We subjected the original list to an updated search in UniProt and retrieved 978 current UniProt entries, including GPCRs, provided in **S1 File.** The second was generated by using three alpha helix prediction tools followed by filtering of splice variants and clustering of the remaining genes [47]. This list contains a total of 3,399 genes, including GPCRs, and is available from their supplement. The mapped *P. damicornis* entries (see below) are available in **S2 File.**

**(ii) Human GPCR list.** A comprehensive list of human GPCRs was extracted from an updated UniProt list of multi-species GPCRs (release: 2020_03 of 17-Jun-2020: 825 proteins (https://www.uniprot.org/docs/7tmrlist.txt). This list contained a total of 3093 GPCR sequences including 825 human GPCRs. See **S3 File** for the extracted human GPCR sequences.

The mapped *P. damicornis* entries (see below) are provided in **S4 File.** Note that this file only contains the top ranked HHblits hit in each case for clarity. In each search, there are usually multiple highly ranked hits and they have been included in the discussion of this paper where applicable.

**(iii) G protein list.** Sequences of human G proteins were obtained from UniProt using keyword search, resulting in 16 alpha chains, 5 beta chains, and 12 gamma chains [30]. The complete list of these human proteins and their candidate *P. damicornis* homologues is provided in **S5 File.**

**(iv) Toll-like receptor list.** Protein sequences of TLRs of human [31] and other organisms, namely Drosophila [48], chicken [49], frog [50], zebrafish [51] and *C. elegans*, were retrieved from UniProt. TLR downstream signaling molecules (MyD88, TIRAM, TIRAP and TRIF) were also acquired from UniProt. The complete list with all the protein sequences used is provided as **S6 File**. The extracted *P. damicornis* TLR list is provided as **S7 File** (note there is a separate sheet for each organism).

## (b) Remote homology detection using Hhblits

To enable organism specific searches, we created an online HHblits coral protein remote homology search tool (https://hhblits.cs.tufts.edu/), an installation of HHblits [18] with genomic databases built for *P. damicornis* and separately for *C. goreaui*. Individual FASTA sequences were imported to this search tool and queries were run with an E-value cutoff of $10^{-3}$, single iteration, minimum probability of 20 (default), and with the minimum number of lines to show in the hit list expanded from 10 to 250. The jobID, email information, database information (e.g. *P. damicornis*) were submitted and the result output file was received by email or as batch predictions. Individual predictions contained the HHblits results of the submitted proteins in text format including protein sequence alignment, E-value, P-value, probability, column matched and score. In the case of GPCR, the result output was analyzed in a bidirectional fashion as follows. First, the top ranked *P. damicornis* hit was retrieved for each GPCR. Then a list of unique *P. damicornis* proteins were created from that, and the corresponding human GPCRs for which they appeared as top hits, provided as **S4 File.** For membrane receptors, the non-redundant lists of top ranked *P. damicornis* hits retrieved in the same way as for GPCRs are provided for both lists (**S2 File**). In the case of TLRs and G proteins, HHblits results were analyzed manually (see Results). For TLRs, top hits with 100% probability score were selected and analyzed separately for each model organism, i.e. human [31], Drosophila [48], chicken [49], frog [50], and zebrafish [51], before comparing them across organisms (Table 1). The list of *P. damicornis* hits for TLRs are provided as **S7 File.** A summary based on the frequency of occurrence of *P. damicornis* hits is provided in **Table 1**.

**Table 1. Number of times a model organism shows homology to a given *P. damicornis* protein with 100% probability.** Columns highlighted in green show at least one representation in each model organism studied.

| Organisms | *P. damicornis* proteins | | | | | | | | | | | | |
|---|---|---|---|---|---|---|---|---|---|---|---|---|---|
| | 22934 | 22930 | 11599 | 9200 | 14109 | 17966 | 13021 | 15883 | 11734 | 21819 | 737 | 15877 | 9057 |
| **Human** | 5 | 5 | 10 | 7 | 4 | 3 | 3 | 2 | 2 | 2 | 1 | 1 | - |
| **Zebrafish** | 8 | 7 | 11 | 9 | 6 | 6 | 2 | - | - | 2 | 1 | - | - |
| **Frog** | 6 | 5 | 10 | 10 | 4 | 4 | 3 | - | - | - | - | - | 1 |
| **Chicken** | 7 | 7 | 9 | 9 | 2 | 2 | 1 | - | - | - | - | 1 | - |
| **Drosophila** | 9 | 9 | 7 | 9 | 2 | 3 | 2 | - | - | 1 | - | 1 | - |
| *C. elegans* | 1 | - | 1 | 1 | 2 | 4 | - | - | - | - | - | - | - |

### (c) PROSITE analysis

The most highly ranked *P. damicornis* sequences retrieved through HHblits were subjected to PROSITE analysis [52] to identify the presence of conserved domains (https://prosite.expasy.org/). Protein sequences were submitted to the PROSITE user interface and the results were analyzed and grouped by combining the domain schematic provided by PROSITE.

### (d) Transmembrane helix detection

The presence of transmembrane helices was predicted using TMHMM Server v. 2.0 (DTU bioinformatics, Department of Bio and Health Informatics) (https://services.healthtech.dtu.dk/service.php?TMHMM-2.0).

### (e) Homology-modeling

Homology models were generated using Swiss Model (https://swissmodel.expasy.org/), an integrated web-based service dedicated to homology modeling of proteins [53]. We used the target-template alignment function of Swiss Model and provided the reconstruction of the full TLR5 (3J0A) combining crystallographic and cryo-electron microscopy data created by [54] as a template to model *P. damicornis* 9200. We have also modeled our potential matches for opsins (629, 2270, 12246, and 19775) using squid rhodopsin as a template (2ZIY) [55]. The models were evaluated for their global quality estimate and local quality score as per Swiss Model guidelines. The models were downloaded and analyzed using PyMOL (version-2.3.4, Schrodinger, LLC). These proteins were further structurally analyzed for ligand binding pocket and Ballesteros-Weinstein numbering system. Ligand binding pocket residues in GPCRs were extracted from previous chemogenomic analysis [56].

### (f) Molecular docking studies of retinal with coral opsin homologs

Molecular docking was performed using retinal as the ligand with AutodockTools 1.5.6 [57]. We standardized our docking experiments using squid rhodopsin with the following parameters: center_x = 43.171, center_t = 6.216, center_z = 17.019, size_x = 16, size_y = 22, and size_x = 26. Docking was performed by extraction after aligning the homology model with the squid rhodopsin space coordinates. The 'exhaustiveness' option was set as 32.0. The binding pocket was analyzed using the Biovia discovery suite 2019 v19.1.0.18287 (Dassault Systemes Biovia Corp). Residues involved in the interactions for each model are listed in **S1 Table**.

### (g) Multiple Sequence Alignment of *P. damicornis*

Potential *P. damicornis* members of respective protein families were aligned to each other using MUSCLE [58] (https://www.ebi.ac.uk/Tools/msa/muscle/) to examine if structurally relevant amino acids were conserved across family members. As an example, the alignment of opsin homologues in *P. damicornis* is shown in **S1 Fig**.

### (h) D-SCRIPT analysis

We initially identified candidate homologs in *P. damicornis* for the human alpha, beta, and gamma G proteins using HHBlits [18] (hhblits.cs.tufts.edu). We took the union of top hits to identify 124 candidate alpha proteins, 207 candidate beta proteins, and 5 candidate gamma proteins. We used the human pre-trained D-SCRIPT [59] model to predict interaction between all pairs of alpha-beta, beta-gamma, and alpha-gamma subunits. We performed the same analysis in *Montipora capitata [60]*, where we identified 184 candidate alpha proteins, 253 candidate beta proteins, and 4 candidate gamma proteins. We created a mapping between

*P. damicornis* and *M. capitata* proteins using BLAST [10] and identified best-bidirectional-hits, i.e. a pair *(P,M)* map to each other if *M* is the best BLAST hit in *M. capitata* for *P* and *vice versa*. We overlay the *P. damicornis* and *M. capitata* networks with each other using the mapping and identify as network-evidence candidate proteins where pairwise interaction is predicted between an alpha, beta, and gamma subunit forming a triangle in the network.

# Results

## (a) Development of a pipeline: Addressing the challenges with existing methods

An overview of the pipeline we developed to address the goal of identifying the repertoire of membrane receptors in the non-model organism, the coral *P. damicornis*, using known information from the model organism, human, is shown in **Fig 2**. The first step is to create a list of human proteins representing the function of interest, here membrane receptors. A list of membrane receptors in human had been published previously [46], but when retrieving the sequences from the UniProt database, several entries were no longer valid. We manually retrieved the updated UniProt ID's by searching the database through protein names. The list of human membrane receptors obtained is available as **S1 File**. This list contains 978 human proteins. We also used another published list of membrane receptors [47], which contained 1352 human proteins reported using an outdated identifier format. Finally, a list of human GPCRs was extracted from a GPCR list in any organism from the UniProt database available at (https://www.uniprot.org/docs/7tmrlist.txt).

Initially, we used BLAST, as well as several multiple sequence alignment (MSA) based tools to retrieve *P. damicornis* homologues, but after inspection of the results we concluded that the alignments between human and *P. damicornis* sequences were poor as judged by the number

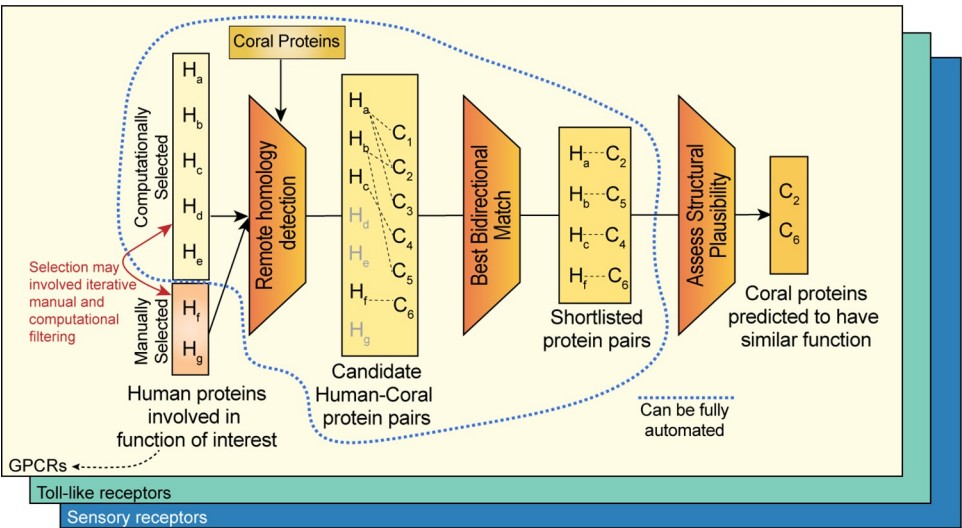

**Fig 2. *In silico* protein prediction methodology.** We employ a multi-step process which combines computational and manual filtering to identify putative functional candidates in coral. For a protein family of interest, a small number of human proteins are manually selected, and a larger number are computationally selected. We perform two remote homology database searches using HHblits—the manually selected human proteins against the database of coral proteins, and the coral hits against the human protein database. We consider a coral protein a candidate only if it is a best-bidirectional match—if the human protein it matched with was the highest likelihood hit in both directions of the search. The process to this point can be fully automated. However, further verification of the coral candidates requires domain-specific knowledge and involves an in-depth analysis of the structural viability of the candidate through both traditional (see Figs 3 and 4) and machine learning (Figs 5 and 6) methods.

of gaps and the fractions of sequences aligned (data not shown). We concluded that due to the low sequence similarity between human and *P. damicornis*, we required a remote homology detection tool. Hhblits is the accepted gold standard for remote homology detection [18]. Thus, all three human membrane receptor lists were searched against *P. damicornis* using our implementation of HHblits, available at https://hhblits.cs.tufts.edu/, where our implementation of HHblits contains genome-wide template libraries of HMM models of all the genes in several coral hosts and symbiont genomes (principally *P. damicornis*, *M. capitata*, plus the clade C1 symbiont).

When mapping these three sets of lists to the *P. damicornis* sequences using HHblits, it became clear that the same genes mapped to multiple human query sequences. This is because there are many members of membrane receptor superfamilies, e.g. receptor tyrosine kinases or GPCRs in each organism itself already. We therefore created non-redundant lists of the top ranked *P. damicornis* sequences. We obtained 374 unique *P. damicornis* sequences from the 978 human plasma membrane receptome (older list, based on [46]), 329 from the 1352 human membrane proteome (newer list, based on [47]) and 151 from the 825 human GPCRs (based on (https://www.uniprot.org/docs/7tmrlist.txt)). These lists and their respective overlap is provided in **S2 File**, labeled "old membrane", "new membrane" and "GPCR", respectively. Combining the three lists and eliminating duplicates appearing in more than one of the lists yielded a total of 446 unique *P. damicornis* sequences. While all 151 GPCRs were found in both membrane receptor lists, there were 111 and 50 non-GPCR receptors missed if we considered only the older human receptor list as compared to the newer one. Thus, we conclude that there are 295 non-GPCR and 151 GPCR candidates in the *P. damicornis* membrane receptome.

It is important to note that, in the majority of cases, none of the quantitative parameters of the sequence alignment was able to differentiate between the hits. In other words, it was not possible to automatically assign a best human homologue for any given *P. damicornis* sequence. This is because different members of protein families have diverged in humans, and are more similar to each other, than they are to any *P. damicornis* sequence, which is evolutionarily most distant to all of them.

Given the fact that many more human sequences map to a smaller set of sequences in *P. damicornis* raises the question which functional sub-categories of these superfamilies are present in *P. damicornis*. Clearly, sequence alone is not sufficient to answer this question, and therefore identifying which of the functionalities of a given sub-family of membrane receptors is present in corals requires an analysis of known functional properties. This requires domain expertise and can no longer be fully automated in contrast to all previous steps (**Fig 2**). This final step of the pipeline is demonstrated for two examples below and necessarily branches according to the functions of the proteins of interest. The opsin subfamily was chosen as a representative example for the type II membrane receptor, the GPCR pipeline (section (b) below). The TLRs were chosen to represent the type I membrane receptors (section (c), below).

## (b) GPCR (type II receptor) pipeline branch

**(i) Global analysis of GPCR families.** The large GPCR family has been divided into subclasses based on a combination of pharmacological and sequence considerations, a classification which has been revised a number of times over the years [56]. Here, we use the clusters obtained with a chemogenomics approach based on the alignment of 30 critical GPCR positions supposed to face the ligand binding cavity [56]. Major classes besides olfactory receptors are Frizzled, Glutamate, Secretin and Adhesion families with the Class A Rhodopsin family being split into 18 different clusters. When analyzing the human receptors for which *P. damicornis* sequences were found, these included chemokine, taste, glutamate, adrenergic, lipids, peptides, adenosine, amines, melanocortins, acids, chemoattractants, purines, frizzled,

adhesion, prostanoids, and MAS-related receptors. Likely missing are melatonin receptors (with 2 low ranked options), vasopeptides, brain-gut peptides, SREB, secretin, opioid and glycoproteins. In all cases there are multiple human sequences that map to multiple (but less than humans) *P. damicornis* sequences. Detailed below is the evidence that *P. damicornis* corals have sensory receptors analogous with smell, taste and light perception in sections (ii), (iii) and (iv), respectively.

**(ii) Odorant receptors in *P. damicornis*.** We found 208 human olfactory receptors in the list of 825 human GPCRs that map to unique pdam genes 17423, 20860, 17300, 5244, 5376 and 16463. This suggests that there are at least 6 olfactory receptors in *P. damicornis*.

**(iii) Taste receptors in *P. damicornis*.** HHblits suggests that there are 15 remote homologs for the human taste receptors in the *P. damicornis* proteome: 04028, 09436, 00629, 02659, 13619, 21435, 22798, 17219, 01145, 13621, 10275, 04281, 02512, 20500, and 16973. 02512, and 09436 have a large N-terminal domain of approximately 500 amino acids preceding the 7 transmembrane helical GPCR motif. This architecture is similar to what is found in human metabotropic glutamate receptors, where the soluble glutamate binding domain precedes the 7 transmembrane helical motif, but the added domains in the above two *P. damicornis* putative GPCR sequences show no similarity with the human glutamate binding domain.

**(iv) Vision receptors (opsins) in *P. damicornis*.** The HHblits result showed that the 11 members of the human opsin family mapped to 7 *P. damicornis* sequences. However, there were other members of the Class A GPCR family that also mapped to the same *P. damicornis* sequences. To find out which of the HHblits remote homology predictions were possiblerepresentations of visual functions, we used the structure-function paradigm and analyzed the ligand binding pockets of these proteins, and the results are displayed in **Figs 3 and 4**. Light detection is the main function of rhodopsin. Therefore, the homologous proteins must have an ability to bind with a light sensitive retinal ligand and should possess a conserved ligand binding pocket. This pocket should show sequence similarity with human opsin protein. Considering this hypothesis, we analyzed the human opsin ligand binding pocket and compared our analysis with *P. damicornis* opsin homologues. We obtained the profiles of 30 cavity-facing amino acids used for clustering of the human GPCRs including opsin [56], shown in **Fig 3A**. Thus, the approach outlined here for opsins can be extended to other GPCRs. The first step was to verify the presence or absence of these amino acids important for ligand binding in the *P. damicornis* sequences for the opsin family. The 30 residues are spread out throughout the sequence, and the Ballesteros-Weinstein numbering scheme used to identify aligned positions in GPCRs clearly showed that these residues are located across the 7 transmembrane helices (**Fig 3A**). We compared these 30 residues against the top hit for OPSD and OPN5, and the result showed that only 11 out of 30 residues are the same between OPSD and their *P. damicornis* homologue.

Further, we had previously determined the minimal ligand binding pocket that is capable of exerting the function of a GPCR: transmission of the ligand binding signal to the downstream signaling proteins [61]. The approach used GREMLIN, Generative Regularized Models of Proteins, to identify longnge interactions from co-varying amino acid positions in multiple sequence alignments [62]. GREMLIN learns an undirected probabilistic graphical model known as a Markov Random Field (MRF). Unlike HMMs, which are also graphical models, MRFs can model longnge couplings (non-sequential residues). We performed GREMLIN analysis on GPCRs, statistically evaluated different sizes of ligand binding pockets and found that a pocket as small as 4 residues still shows significant enrichment of edges over null. This means that four residues connect maximally to the rest of the protein orchestrating the global conformational change inherent to GPCR function and are required for signal transduction. Our analysis showed that 3 of these 4 residues are conserved across human opsin and the *P. damicornis* homologue. This supports the conclusion that this protein is a functional GPCR.

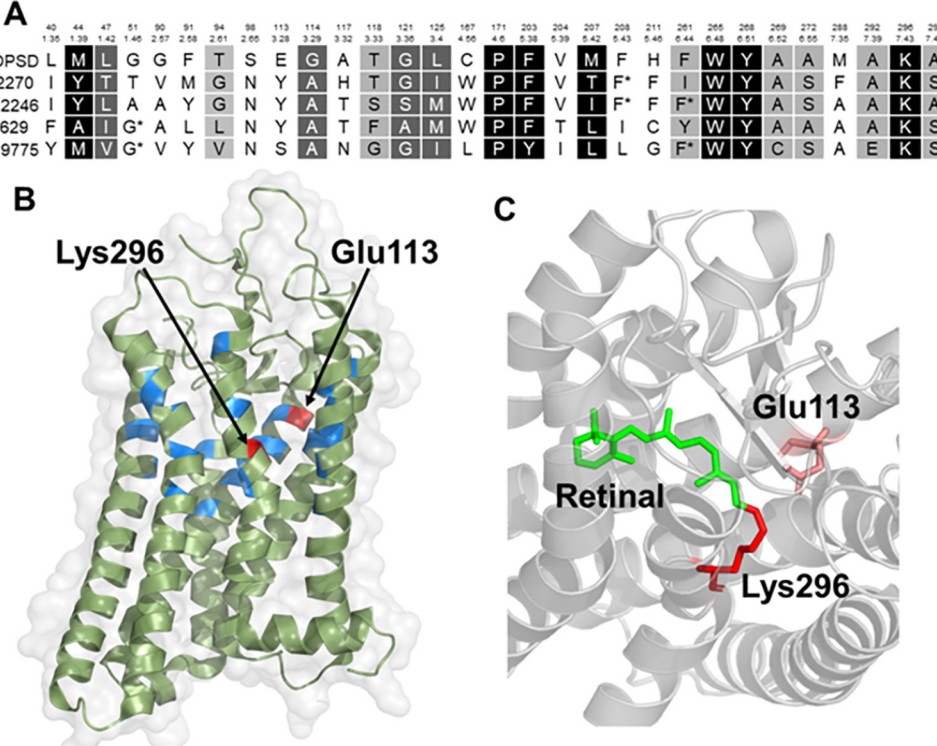

**Fig 3. Sequence alignment and active site analysis of rhodopsin.** (A) Alignment of human rhodopsin with remotely homologous coral proteins, only showing the positions of those amino acids that are in proximity to the ligand in the opsin family (ref). TheBallesteros-Weinstein numbering scheme is used to enable comparison to other members of the GPCR family. Positions are colored according to the degree of similarity: White foreground/black background, 100%; white foreground/grey background, > 80%, black foreground/grey background, > 60%. (B) Active site of bovine rhodopsin crystal structure 1L9H (>99% homologous to human rhodopsin) showing the residues involved in the active site shown in (A). (C) Active site showing the interaction of the ligand and vitamin A derivative, retinal, which makes rhodopsin light-sensitive, with Lys296 and Glu113 (also labeled in A,B).

Finally, GPCRs involved in vision require a lysine in a specific location of the ligand binding pocket to covalently attach via a Schiff base. The Schiff base is stabilized by another residue in the sequence, that is glutamic acid (E113), forming a counter ion to lysine K296, (highlighted by red arrows in **Fig 3B**). We can see that the *P. damicornis* homologue contains the lysine required for retinal binding, but has a tyrosine instead of glutamic acid in the counter ion position. This substitution is expected to cause a shift in the absorbance maximum but not a loss of function as a light-sensitive protein. This finding supports the hypothesis that this *P. damicornis* sequence represents a light-sensitive opsin protein, likely with an absorbance maximum different from that of rhodopsin and more like the OPN5 protein, which also has a tyrosine at this crucial position. Repeating this process for all of the *P. damicornis* homologues allowed us to eliminate proteins that did not fit the requirements of GPCR action or retinal binding ability and narrow down the most closely related opsin protein. We predict that there are 4 *P. damicornis* opsin proteins (in contrast to 11 human opsin proteins) and these are most similar to OPN5, OPSG, OPSX and OPSR.

**(v) Homology modeling and molecular docking studies with opsin homologs.** To confirm the structure and functional relationship of rhodopsin with putative coral rhodoposin homologs, we have performed structural modeling of the top coral rhodopsin hits using the squid rhodopsin structure as a template (**Fig 4**). We carried out molecular docking to these

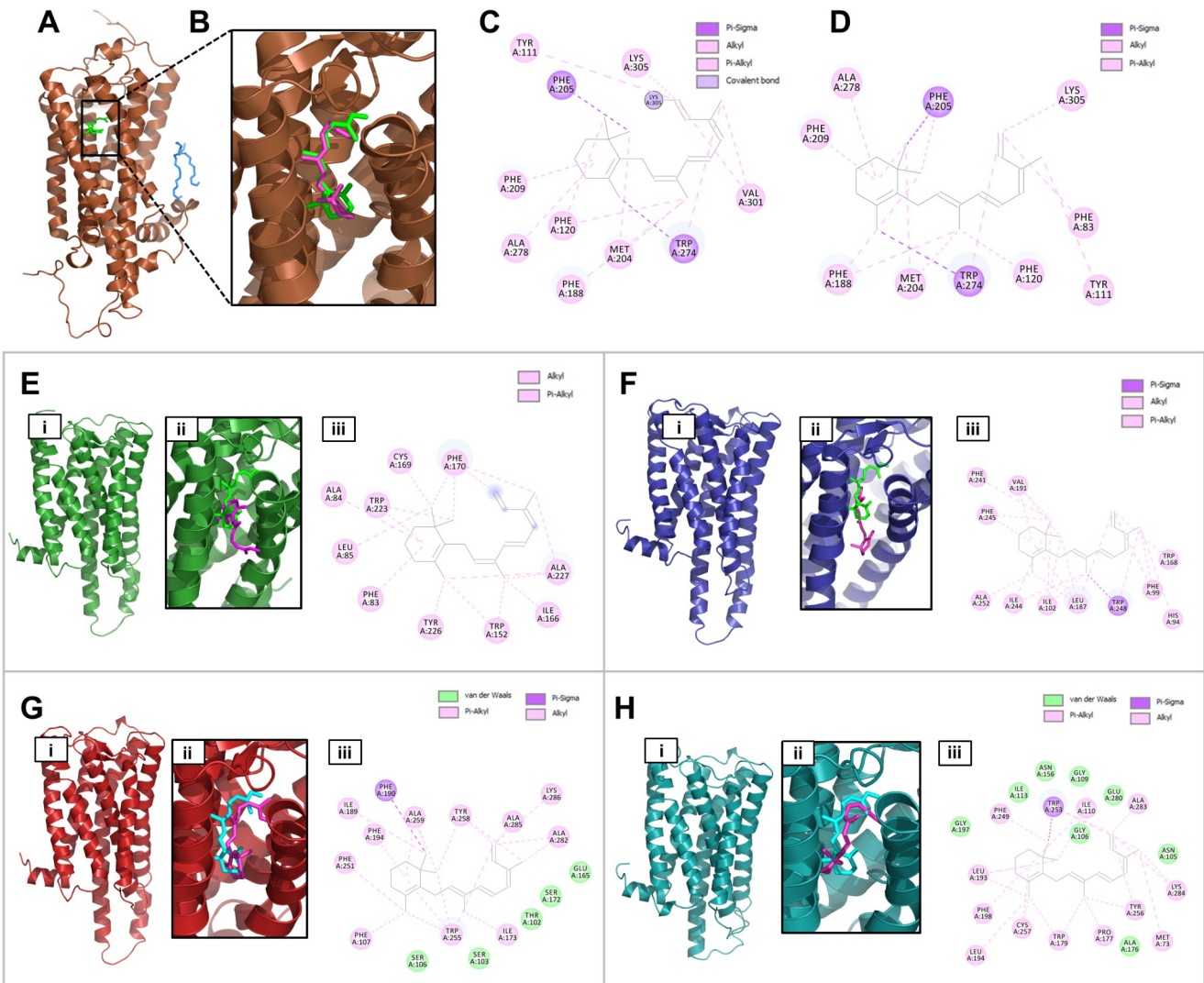

**Fig 4. Do the putative coral opsin proteins have a propensity to bind retinal which would make them light sensitive? To address this question, we created homology models and conducted molecular docking of the "rhodopsin" receptors with retinal.** (A) Crystal structure of squid rhodopsin (2ZIY). Squid is the evolutionarily closest organism to corals of those for which rhodopsin structures exist. (B) Expanded view of the retinal (shown in green in all structures A-H) in the squid crystal structure shown in (A). As a control, the same retinal was docked to the structure and the docked conformation is shown in pink. (C, D) Molecular interactions of retinal in the crystal structure vs. docked conformation. (E, F, G, H) Homology modeling and docking studies with coral homology models 629, 2270, 12246 and 19775, respectively. In all cases, panels are labeled as follows: (i) homology model, (ii) retinal confirmations: Squid(green) vs. docking pose in coral opsin (magenta), (iii) molecular interaction of docking pose. Details of interactions and the docking scores are provided in **S1 Table**.

homology models that suggested that the retinal was indeed able to bind with the putative ligand binding pocket in the coral rhodopsin homologous structures. This suggests that these are strong candidates to be considered as coral rhodopsin homologs and may play an important role in light sensing mechanisms. We have summarized the details of the retinal interactions with squid rhodopsin and coral rhodopsin homologs in **S1 Table.**

## (c) GPCR binding partner: G Proteins

To gather further evidence that the proteins identified were opsins, we investigated their signal transduction function, namely binding and activation of the G proteins upon ligand binding

(or light activation of the retinal ligand in the case of opsins). G proteins are heterotrimeric proteins consisting of alpha-, beta- and gamma-domains. There are 16 different alpha chains, 5 beta chains, and 12 gamma chains in humans [30], and their UniProt ID's are provided in **S5 File, tab 1**. HHblits search of the alpha and gamma chains yielded six potential *P. damicornis* alpha chains and five potential gamma chains, see **S5 File, tab 2**. We were unable to retrieve potential beta chains reliably, due to the beta propeller structure of the beta chains being a very common structural motif found across multiple protein families.

**(i) Multiple sequence alignment of *P. damicornis* alpha chains**. The potential *P. damicornis* alpha chains were aligned with the human G-alpha subunit that binds to rhodopsin, called transducin, with gene symbol GNAT1 using MUSCLE (see Methods) to examine if structurally relevant amino acids were conserved. Amino acids of GNAT1 which interact with rhodopsin, GNB1, and GTP were identified from co-crystal structures (PDB ID: 6OY9, 1TAD) by selecting residues within 5 Å of the relevant structure in PyMOL. These residues are highlighted in **Fig 5A** using boxes colored orange, blue, and green, respectively, corresponding to the structures shown in **Fig 5B and 5C**. The alignment shows that the GTP binding pocket residues are the most conserved, the receptor binding residues are the least conserved, and the beta chain binding residues are more conserved than the receptor binding residues but less conserved than the GTP binding residues. This is to be expected, as the GTP pocket will need to bind to the same structure in corals as it does in humans, whereas the receptor and beta chain binding residues will need to bind to coral receptor and beta chain homologues, which will be structurally different. Because the conservation patterns are similar across all 5 *P. damicornis* G-alpha candidates, we are not able to select a top candidate from this pool that is most likely representative of the rhodopsin-binding G-alpha subunit, transducin (GNAT1).

**(ii) Network analysis of putative *P. damicornis* G protein subunits**. To overcome the challenge that sequence alignment alone was not sufficient to narrow down the choices of *P. damicornis* sequences related to G proteins, we leveraged computational prediction of protein-protein interactions (PPI) to characterize the relative likelihood of the top *P. damicornis* hits being truly involved in GPCR binding activity (**Fig 5D and 5E**). Applying D-SCRIPT [59], a recently introduced sequence-based deep learning model for PPI prediction, we performed two complementary PPI analyses. In the first, we performed an all-vs-all computational screen of interaction between all the candidate G-alpha, beta and gamma proteins in *P. damicornis*, reasoning that the true positive hits will display the expected interaction patterns of alpha-beta and beta-gamma binding. Our predicted PPIs broadly conformed to this expectation, with 17900, 11071, 23984, 07710, 14456, 11840 being the strongest candidates for alpha subunits, 00168 the strongest candidate for a beta subunit, and 00526 the strongest candidate for a gamma protein (see **Fig 5D** and **S5 File, tab 2**). To gain further confidence in our estimates, we also performed a similar screen of G protein subunits in *M. capitata* (**Fig 5E**) and assessed if homologous pairs across the two coral species (estimated by a bidirectional best hit analysis) had similar PPI patterns (see Methods). These results further supported our estimate, as the same connectivity patterns observed in *P. damicornis* are also observed between homologues in *M. capitata* (**Fig 5E**).

**(ii) Conservation of structural interfaces in G protein complexes.** To further confirm the predicted G protein complex compositions, D-SCRIPT was then used to assess the structural plausibility of individual G-alpha candidates in *P. damicornis*. We performed an *in silico* mutagenesis study of each G-alpha candidate, evaluating these mutations by how they

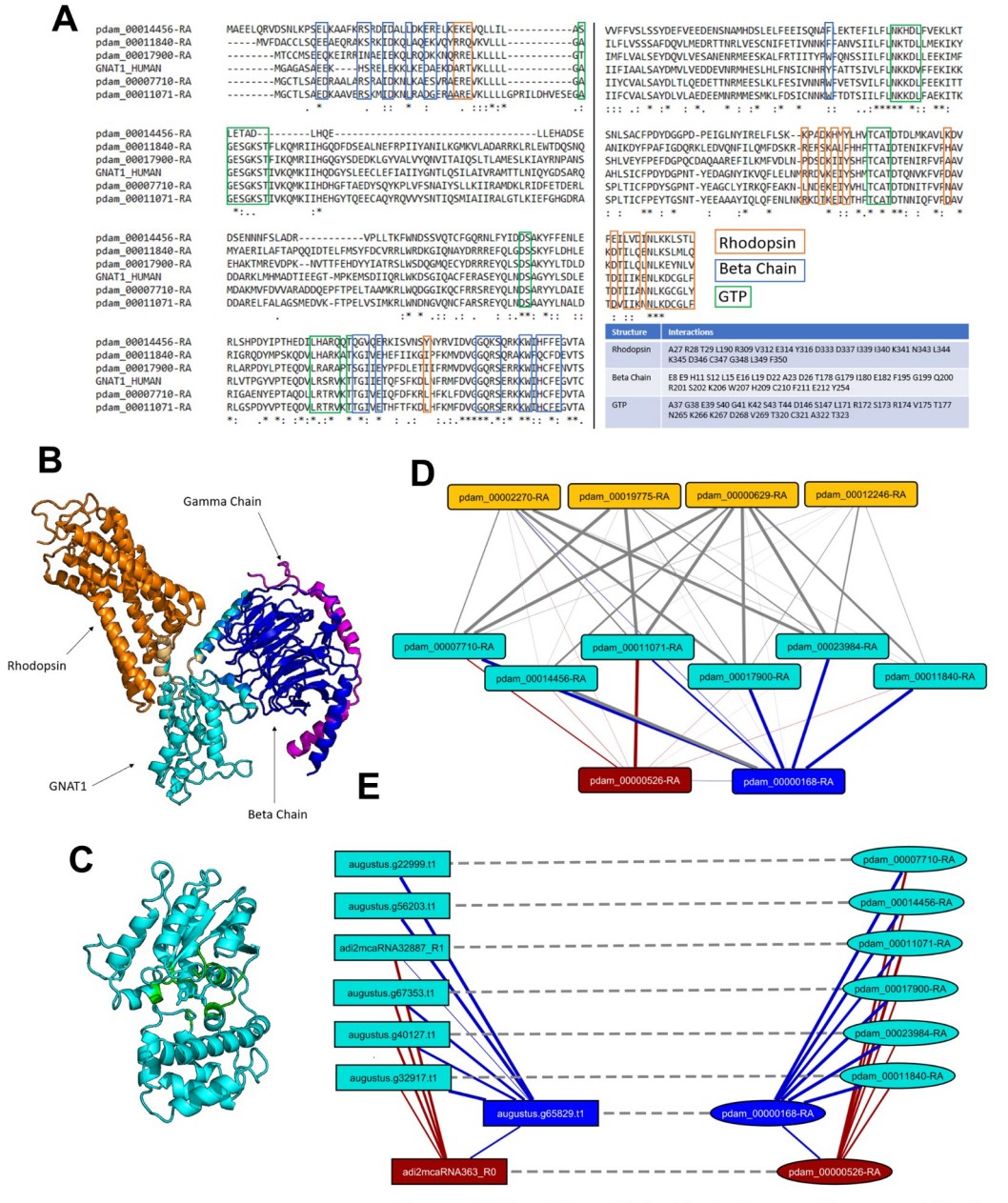

**Fig 5. Identification of putative G protein subunits as candidates of immediate downstream signaling partners for GPCRs.** (A) MSA of GNAT1 with coral homologues and crystal structures of GNAT1. Amino acids of GNAT1 which interact with rhodopsin, GNB1 (beta chain), and GTP are boxed in blue, red, and green, respectively. The same amino acids are also listed in the table. In the crystal structures, GNAT1 (cyan) binding interfaces with rhodopsin (orange) and the beta chain (blue) are shown in (B) and the GTP binding pocket (green) is shown in (C). (D) Predicted interactions in *P. damicornis* between alpha (cyan), beta (blue), gamma (orange) chains and rhodopsin (orange). (E) Predicted candidate G protein interactions in *P. damicornis*, and the predicted interactions of the corresponding candidates in *M. capitata* (solid line: Predicted interaction; dashed line: Best-bidirectional BLAST hit).

changed the D-SCRIPT score for interaction with candidate G-beta proteins 00168 and 14586 (**Fig 6**). Reasoning that a conservative test would be to require the same binding mechanism as seen in human G-alpha-beta interaction, we first aligned candidate G-alpha protein sequences in *P. damicornis* against human G-alpha proteins GNAT1, and used the

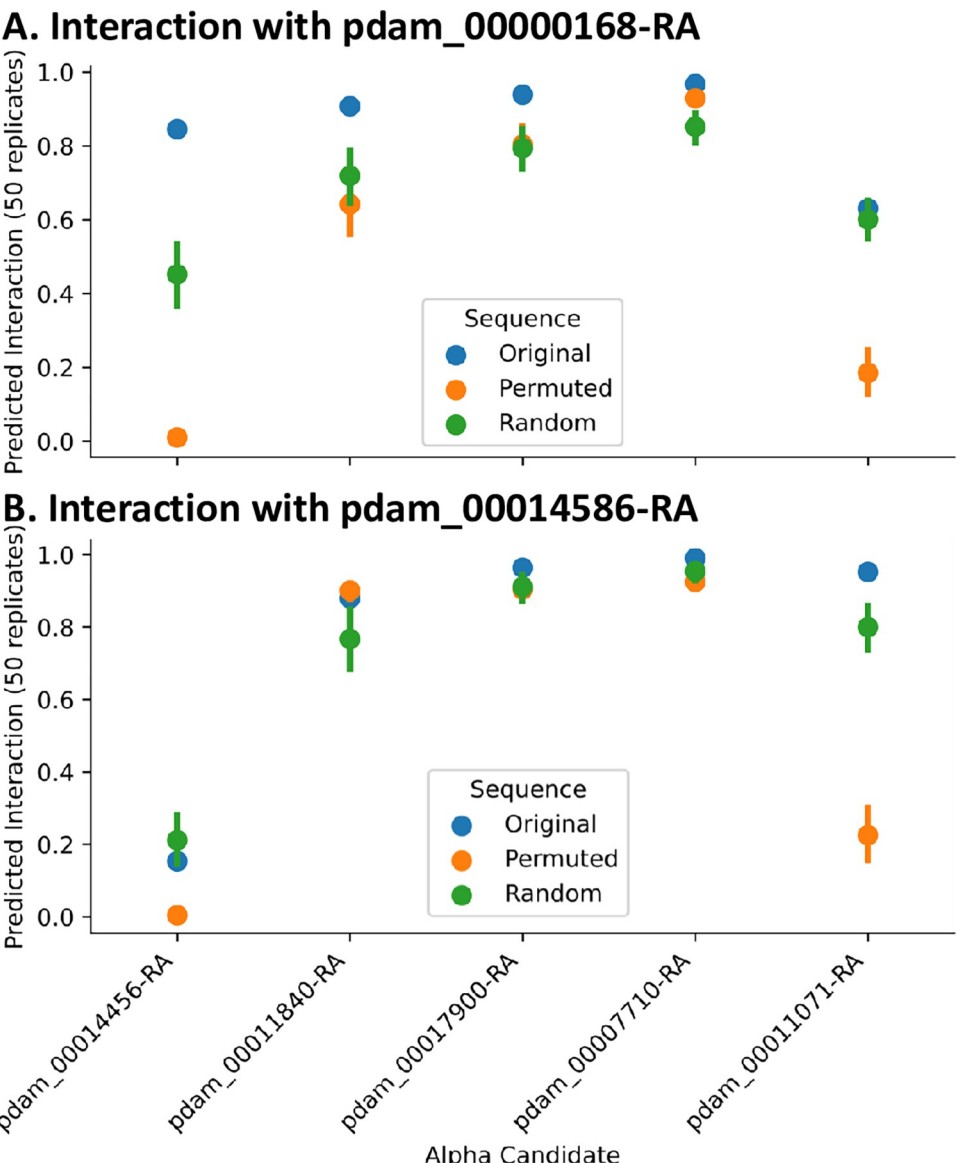

**Fig 6.** Predicted interaction of candidate alpha proteins with beta candidate proteins 00168 (A), 14586 (B). *Original*: Original predicted probability. *Permuted*: Average predicted probability of 50 samples with 25 beta-binding residues randomly perturbed. *Random*: Average predicted probability of 50 samples with 25 randomly chosen residues randomly perturbed.

multiple sequence alignment to identify the residues in coral G-alpha where the corresponding location in GNAT1 is known to play a role in binding to G-beta proteins. We tested if random *in silico* mutations at these locations had a particularly deleterious effect on the likelihood of PPIs with coral G-beta candidates (average of 50 trials). We compared these likelihoods to the original predicted probability of interaction, and the predicted probability of interaction after *in silico* mutations at random sites (average of 50 trials). Two candidates, 14456 and 11071, showed sharply decreased likelihood of a PPI occurring specifically when the sequences were mutated at putative binding locations—suggesting a binding mechanism similar to human G-alpha-beta interaction is conserved in determining the interaction likelihood of coral G-alpha-betas.

### (d) Toll-like receptor (TLR, type I receptor) pipeline branch

**(i) TLRs in *P. damicornis*.** After our investigation of putative GPCR (aka type II receptor) proteins and pathways in *P. damicornis*, we next investigated a representative of the type I receptors, namely TLRs. Because there are large differences in the numbers of TLRs in different organisms, we retrieved TLR sequences not only from humans [31], but also zebrafish [31,51], frog [50], chicken [49] and Drosophila (Toll proteins) [63]. After subjecting these sequences to HHblits prediction, we extracted all coral proteins that showed a minimum of 100% probability, and grouped them by their homology to a given organism. The results are summarized in **Table 1**. *P. damicornis* proteins 22934, 22930, 11599, 9200, 14109, 17966, and 13021 were observed as homologues to at least one TLR from each of the five model organisms studied. Because many type I receptors are often composed of multiple domains, we then subjected these proteins to PROSITE analysis in addition to homology modeling.

**(iii) PROSITE analysis showed similar domain signatures in *P. damicornis* proteins.** PROSITE analysis was used to identify domain structures within putative TLRs. **Fig 7A** illustrates that in human TLRs, there are multiple copies of leucine rich repeat (LRR) domains present on the extracellular side of the receptor and a Toll/interleukin-1 (IL-1) receptor (TIR) domain in the intracellular side in human TLRs. In contrast, *P. damicornis* 22934, 22930, 15883, 15877, 11734, and 11599 were devoid of LRRs and only displayed the TIR domain (**Fig 7B**). This suggests that these proteins may not belong to the typical TLR family. On the other hand, PROSITE analysis of *P. damicornis* 14109 and 17966 homologues showed a large number of LRRs, plus a cadherin domain, EGF_CA (Calcium-binding EGF domain), Thrombospondin, type 3 repeat (TSP3), Thrombospondin C-terminal domain profile (TSP_CTER) domains on the extracellular side and the TIR domain on the intracellular side. Because of these additional domains, it is possible that these are TLRs but with different or additional functions as compared to their endosomal human counterparts. A third domain composition was revealed by PROSITE analysis of *P. damicornis* 737 which only had LRRs and no TIR domain. Therefore, this homologue was rejected as a TLR candidate. The only domain composition similar to that of human TLRs was observed for *P. damicornis* homologue 9200, which included multiple LRRs in the extracellular domain and a TIR domain in the cytoplasmic domain. For further analysis, we selected 9200 for homology modeling due to its matching profile with human TLRs.

**(iii) Structural similarities between human TLR5 and *P. damicornis* TLR.** Swiss Model matching for query *P. damicornis* protein 9200 retrieved human TLR5 (3j0a) as the top hit. We refined the model by removing several missing regions from the coral TLR model (**Fig 8**). Using TMHMM with 9200, we identified a transmembrane helix region (649–671) between the extracellular and the intracellular region, supporting the organization as a typical type I receptor.

**(iv) MyD88 homologue as the possible downstream partner**. TLRs utilize multiple adapter proteins to transmit the signal to the inside of the cell, namely MyD88, TIRAM, TIRAP and TRIF adaptor proteins. Our HHblits results of these proteins showed presence of a coral homologue only for MyD88.

## Discussion

We present a general pipeline applicable to any non-model organism to explore structures and functions based on detectable homology to critical proteins in humans or other model

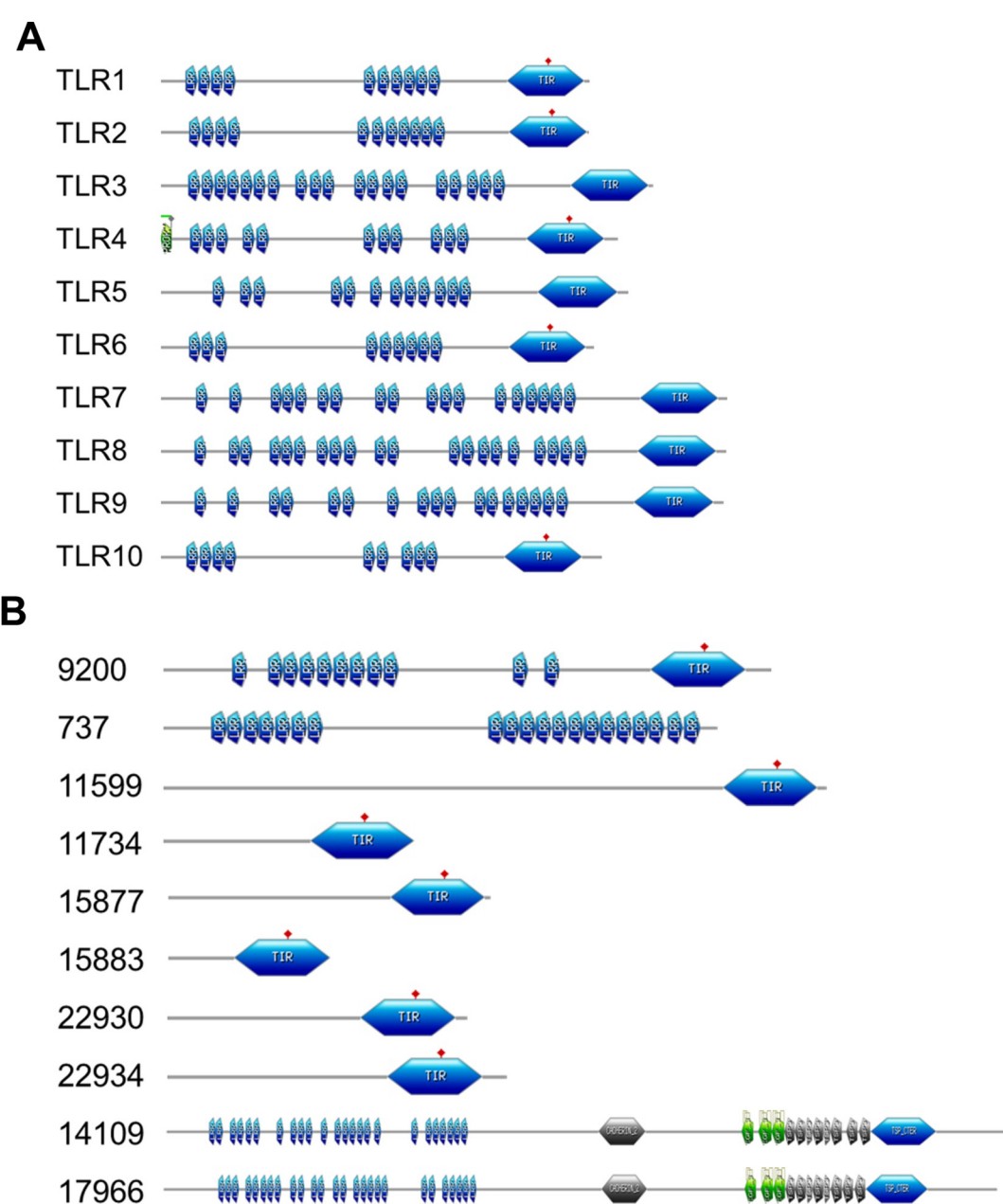

**Fig 7. PROSITE analysis of human and coral TLRs.** (A) PROSITE domain mapping of human TLRs. (B) PROSITE domain mapping of possible *P. damicornis* TLR homologues.

organisms (**Fig 2**). Using three case studies, we demonstrate customized, structure-based approaches to select functional candidates from the pool of homologs. In the case of GPCRs, we looked at specificity determining positions [64,65], which can then be confirmed using 3D structural modeling, to check that the active sites and binding pockets that are expected, should the functional role of the protein be conserved, are indeed present. In the case of G proteins, we derived function based on protein-protein interactions, where we have used our recent deep learning method [59] to perform *in silico* mutagenesis studies to further help us distinguish sequences which are likely to allow us to correctly transfer functional annotation

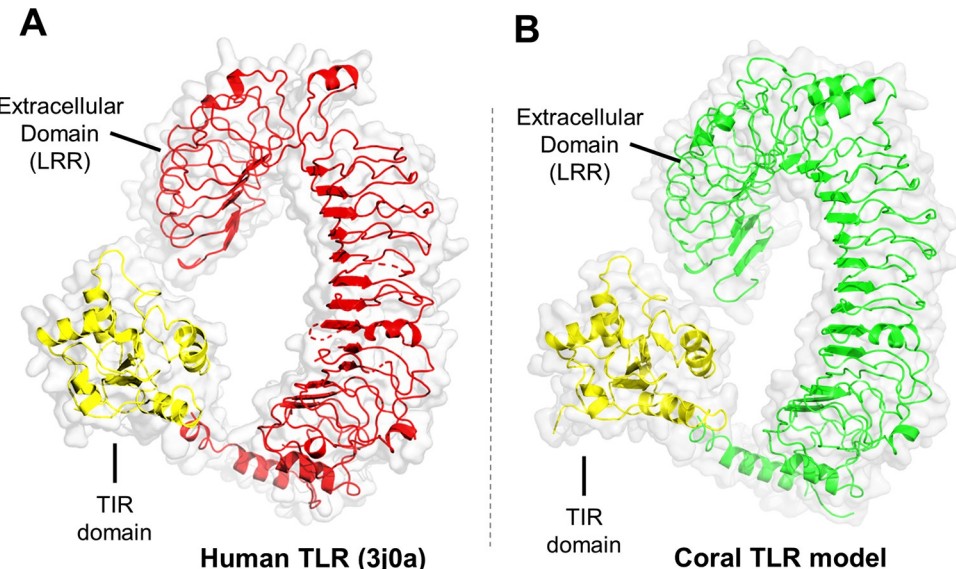

**Fig 8. Coral TLR homology model based on the full-length TLR5 structure model with pdb id 3j0a LRR: Leucine rich repeats.** (A) Structure reconstruction of the full-length human TLR5 protein (3j0a). (B) Homology model of *P. damicornis* 9200 protein using the human TLR5 structure reconstruction model as a template.

from their human homologs from those that may have other functions. In the case of TLRs, we have used PROSITE to identify the presence or absence of entire domains within the sequences, and used expert knowledge to evaluate if the location of these domains matches the functional expectation (e.g. if the domain faces the inside or outside of the cell).

Generally, the availability of three-dimensional structures for one or more members of the protein family of interest opens the possibility for in-depth structure modeling using homology. Although the quality of homology models depends on the degree of sequence conservation, structurally aligning remote sequences goes a long way in interpretation of their structure conservation as long as we can have confidence in the alignment. For example, despite near negligible sequence conservation between rhodopsin and metabotropic glutamate receptors, the most distant members of the GPCR family, we were able to predict the pharmacological outcome of ligand binding based on structural alignment [66]. With the continuous improvements in *de novo* structure prediction [67], this task will become more and more feasible even for proteins that do not have any homologous structures available, i.e. for which no structural template can be found.

One challenge not currently addressed by our pipeline is the identification of *de novo* proteins with *de novo* functions. While homology modeling allows us to detect proteins that are present (with mutations) in both species, we acknowledge that just as some proteins novelly evolved in vertebrates and are not present in cnidarians, there is the possibility of complexes that novelly evolved in corals, which this approach cannot detect.

The results of our study have several biological implications. Cnidarians as evolutionarily early animals offer great opportunities to study the evolution of important biological functions, such as sensing and signaling, and understanding host-microbe communication.

TLRs are used for molecular communication with microbes. These receptors interact with specific microbial antigens and play a major role in innate immunity. We obtained strong evidence for TLR presence in *P. damicornis*, in line with several previous reports suggesting the presence of TLR mediated signaling in corals. Analysis of the *Nematostella vectensis* genome

(sea anemone, cnidarian model system) indicated the presence of immune related genes, membrane attack proteins, complement pathway associated signaling molecules [68,69] and one TLR associated with stinging cells (cnidocytes [44,70,71]. RNA-Seq data also supports the ability for immune responses in corals [44,70]. One study showed that the muramyl dipeptide (MDP), a bacterial cell wall component, was found to trigger the up-regulation of GTPases of immunity-associated proteins in *Acropora millepora* [72]. Finally, the transcriptomic expression during pathogen challenge with *Vibrio sp.* in *Acropora cervicornis* showed two TLR2 homologs and the adaptor molecule TNF receptor-associated factor 3 (TRAF3) were up-regulated [73,74]. We now add detailed protein structural analysis to these reports, and also provide evidence that *P. damicornis* carries the gene for the TLR adapter protein, MyD88 (*P. damicornis* id 15711).

Of particular note is that our results suggest a much smaller number of proteins involved in TLR signaling than in the model organisms human, chicken, zebrafish, frog, and Drosophila. We predict that there may only be a single, unique TLR homolog in the coral *P. damicornis* that exhibits all the features expected for TLRs, and is most similar to TLR5 in human which could be trait common to all Anthozoa, not just *P. damicornis* and *N. vectensis*. Furthermore, we have found only one (MyD88) of the four known adapter proteins MyD88, TIRAM, TIRAP and TRIF. This suggests the presence of a less diversified and simpler TLR signaling pathway in coral as compared to higher eukaryotes.

A similar conclusion was reached in the analysis of the large and diverse GPCR family. The human genome alone encodes 825 GPCRs (https://www.uniprot.org/docs/7tmrlist.txt), while we only found 151 GPCRs in *P. damicornis*. Looking at reports of GPCR family size in evolutionarily early organisms suggest that there is a dramatic expansion in the number of GPCRs when transitioning from unicellular to multicellular organisms and the development of sensing-specific organs [75]. For example, choanoflagellates have only 10 GPCR [76], while holothurians, one of the five Echinoderm classes, have 246 GPCRs [77], sea urchins, also echinoderms, have 979 [78] and the Placozoan *Trichoplax adhaerens* has 420 [79], sponges have 330 [80], *N. vectensis* have 890 [81] and hydra has 1200 (!) [82]. Evolutionary analysis of the sub-families suggests that the doubling that takes place between *T. adhaerens* and *N. vectensis* is roughly maintained and most sub-families remain relatively constant in their size distributions, with the Class A Rhodopsin-like family being the largest and most diverse in all species [83,84]. Invertebrates possess remarkable chemosensory capabilities to explore and detect minute biochemical compounds. Despite the low number of odorant receptors found in P. damicornis, corals have demonstrated high sensitivity to chemical cues (e.g. [85] and we postulate that this low number can be explained by the absence of organs dedicated to smell compared to higher organisms. Cnidarian polyp tentacles serve for prey capture and transport (to oral cavity), aggression and defense, and the cells they are made of (cnidocytes) also have various specialized functions [86] however, they constitute, with the oral cavity and the epidermis, the most specialized cnidarian sensing organ.

Generally, odors are used by organisms for survival purposes, such as homing, finding food, distinguishing between the same species and predators, predator avoidance and defense, and reproduction. In humans, the nose is a specialized organ to detect volatile compounds, and so we usually associate the ability to smell only with land animals. However, marine organisms can also smell. For example, fish have nostrils and pump water through them, and the compounds detected are decoded by their olfactory bulb. Fish can detect very low concentrations of compounds and use it for example to detect the direction of their home reef [87] Anemones have long been known for their production of the pyridinium compound, amphikuemin, that attracts its symbiotic fish which swims towards it [88]. In humans the detection and response to olfactory cues is so important that a remarkable gene expansion akin to that of

antibody diversity has taken place giving rise to hundreds of olfactory receptors, which are highly diverse in different individuals underlying the differences between preferences and detection abilities in human populations [89]. Although corals emit large numbers of volatile compounds [90] for marine organisms, it is less important if compounds are volatile in air, as they would be dissolved in water, and so olfactory reception is better referred to as chemoreception in general. This can be well demonstrated with the sea urchin [78]. For example, genomic clustering and single-exon gene structures suggest rapid gene duplication creating a new class of GPCRs, the "group B surreal-GPCR's in the sea urchin that are differentially expressed in pedicellariae and tube feet, suggesting that these organs take up an analogous role to the human nose in sensing chemical stimuli [78]. A similar rapid gene duplication gave rise to the diversity of olfactory receptors in humans, suggesting a similarity in the benefit of expansion of the organism's capabilities to detect a larger diversity of ligands. These gene duplication events are highly organism specific, for example the holothurians are predicted to contain 57 olfactory receptors [77]. The next step after identification of the receptor repertoire is of course the identification and matching of corresponding ligands. Most progress to date with respect to ligands has been made with neuropeptides [91]. Interestingly, there are larger numbers of putative taste receptors as compared to odorant receptors in *P. damicornis* (15 coral for 28 human taste receptors), which might indicate that sensing chemicals as part of the food intake machinery is an important chemosensory part of *P. damicornis*.

Much simpler than differentiating the diversity of chemical compounds is sensing the color and intensity of light, a task that can be accomplished by humans with 11 members of the opsin subfamily of class A GPCR. Again, *P. damicornis* shows a reduced repertoire: based on our detailed sequence and structural analysis of opsin candidates we predict 4 GPCR that may be able to bind retinal and gain light sensitivity in *P. damicornis*. This is consistent with the hypothesis that *P. damicornis* can likely sense and respond to light and even distinguish colors. Previous research showed that coral larvae preferentially settle on red as opposed to white surfaces [92]. Consistent with this behavioral finding, is the inclusion of the red opsin protein (OPSR) in our list. On the other hand, there is also evidence for the ability to sense blue light— the action spectra of polyp retraction in the corals *E. fastigiata* and *M. cavernosa* have absorbance maxima of 482 and 478 nm, with half width at half height of 115 and 105 nm, respectively [93]. These narrow band widths support the conclusion of a single opsin involved in light sensing for polyp retraction in each coral, consistent with the small number of opsin candidates obtained for *P. damicornis*. The counter-ion of vertebrate opsin E is replaced by Y in invertebrate opsins in general [94], also observed for *P. damicornis*, with the exception of S in 19775. The 19775 protein maps to both medium wave and long wave sensitive human opsins, further suggesting a change in absorbance maxima. Thus, despite the absence of eyes, which are only needed when spatial vision is needed, detection of light is a fundamental ability shared with many other organisms, such as sea urchins [78]. Even when such organs are missing, the expression of specific opsin genes is not ubiquitous, providing a pathway towards encoding spatial information.

Taken together, both TLR and GPCR analysis suggest that *P. damicornis* represents a transition species that carries the minimal number of essential components for signaling and innate immunity, while additional functionality and fine tuning is achieved through diversification of this small pool of proteins in higher evolved organisms. Thus, being able to differentiate between early basic versions of a given function and finding the reasons for the need to diverge emphasizes the utility of this method assuming the best hit is the most functionally relevant hit. This may help to identify possible functions for currently not conclusively annotated proteins in *P. damicornis* and other non-model organisms. Although we restricted our detailed structure-function analysis to the GPCR subfamily of opsins in vision, G proteins, and TLRs,

we also presented some initial analysis of possible odorant, taste and other receptors in corals. It is important to note, however, that the discovery of homologous receptors is done mainly through bioinformatics approaches, and thus can only be viewed as hypothesis generating work. Experimental approaches, such as pharmacological, cell biological and/or biochemical, are necessary to confirm these hypotheses. Thus, while further functional and especially experimental tests are needed to verify these predictions, the approach presented identifies potential new targets for studying coral biology and possibly draw connections between the general functions of the human proteins that the coral proteins mapped to and their potential role in coral sensing.

## Supporting information

**S1 Fig. Clustal omega multiple sequence alignment of opsin homologs.** Residues involved in active site formation are highlighted in blue.
(DOCX)

**S1 Table. The details of residues and their molecular interactions with the retinal active site in squid rhodopsin and coral putative rhodopsin proteins.**
(DOCX)

**S1 File.**
(TXT)

**S2 File.**
(XLSX)

**S3 File.**
(TXT)

**S4 File.**
(XLSX)

**S5 File.**
(XLSX)

**S6 File.**
(XLSX)

**S7 File.**
(XLSX)

## Author Contributions

**Conceptualization:** Rohit Singh, Lenore Cowen, Judith Klein-Seetharaman.

**Data curation:** Roshan Klein-Seetharaman, Noah M. Daniels.

**Funding acquisition:** Hollie Putnam, Jinkyu Yang, Nastassja A. Lewinski, Lenore Cowen, Judith Klein-Seetharaman.

**Investigation:** Lokender Kumar, Nathanael Brenner, Samuel Sledzieski, Liza M. Roger, Bonnie Berger, Hollie Putnam, Jinkyu Yang, Rohit Singh, Noah M. Daniels, Lenore Cowen, Judith Klein-Seetharaman.

**Methodology:** Samuel Sledzieski, Roshan Klein-Seetharaman, Bonnie Berger, Rohit Singh, Noah M. Daniels, Lenore Cowen.

**Project administration:** Lenore Cowen, Judith Klein-Seetharaman.

**Resources:** Hollie Putnam, Lenore Cowen, Judith Klein-Seetharaman.

**Software:** Samuel Sledzieski, Monsurat Olaosebikan, Matthew Lynn-Goin, Roshan Klein-Seetharaman, Rohit Singh, Noah M. Daniels, Lenore Cowen.

**Supervision:** Bonnie Berger, Rohit Singh, Lenore Cowen, Judith Klein-Seetharaman.

**Validation:** Liza M. Roger, Hollie Putnam, Judith Klein-Seetharaman.

**Visualization:** Nathanael Brenner, Samuel Sledzieski, Rohit Singh, Judith Klein-Seetharaman.

**Writing – original draft:** Lokender Kumar, Nathanael Brenner, Samuel Sledzieski, Rohit Singh, Noah M. Daniels, Lenore Cowen, Judith Klein-Seetharaman.

**Writing – review & editing:** Samuel Sledzieski, Liza M. Roger, Roshan Klein-Seetharaman, Bonnie Berger, Hollie Putnam, Nastassja A. Lewinski, Rohit Singh, Noah M. Daniels, Lenore Cowen, Judith Klein-Seetharaman.

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
