## [Decision Letter · Decision Letter 0]

11 Jan 2022

PONE-D-21-34522Transfer of Knowledge from Model Organisms to Evolutionarily Distant Non-Model Organisms: The Coral Pocillopora damicornis Membrane Signaling ReceptomePLOS ONE

Dear Dr. Klein-Seetharaman,

Thank you for submitting your manuscript to PLOS ONE. After careful consideration, we feel that it has merit but does not fully meet PLOS ONE’s publication criteria as it currently stands. Therefore, we invite you to submit a revised version of the manuscript that addresses the points raised during the review process. For example, one referee states that the conclusions, in particular about senses and corresponding signaling cascades are "too subjective" and needs a revision. Apart from over-interpreting the results, the manuscript needs a thorough re-organization as suggested by both referees.

We look forward to receiving your revised manuscript.

Kind regards,

Karl-Wilhelm Koch, Ph.D.

Academic Editor

PLOS ONE

Journal Requirements:

“This work was sponsored by NSF grants HDR: DIRSE-IL 1940169 and RAPID 2031614.”

Reviewers' comments:

Reviewer's Responses to Questions

**Comments to the Author**

1. Is the manuscript technically sound, and do the data support the conclusions?

Reviewer #1: Yes

Reviewer #2: Partly

2. Has the statistical analysis been performed appropriately and rigorously? 

Reviewer #1: Yes

Reviewer #2: N/A

3. Have the authors made all data underlying the findings in their manuscript fully available?

Reviewer #1: Yes

Reviewer #2: Yes

4. Is the manuscript presented in an intelligible fashion and written in standard English?

Reviewer #1: Yes

Reviewer #2: No

5. Review Comments to the Author

Reviewer #1: The article « Transfer of knowledge from model organisms to evolutionarily distant non-model organisms : the coral Pocillopora damicornis membrane signaling receptome » discusses the difficulty in finding, with the genomic resources of model organisms, homologs in model organisms unconventional. The article proposes a bioinformatic approach, clearly detailled, in the form of a manual, which can be used in corals as in other non-model organisms, in order to find receptor homologs. This document opens with a long and rich introduction, written with clarity, thus facilitating comprehension by a non-bioinformatician. However, the legends of the figures could be more explicit in order to improve the reading and the comprehension of the document. In my opinion, the discussion is short compared to the introduction and could be enriched . Despite these remarks, the article deserves to be published with minor corrections. Here are my comments :

Introduction

- Add capital letters to : « Hidden Markov Models » instead « hidden Markov models »

Material and methods

- Supplementary file-S2 I'm not sure I understand what the old and new membrane mean

- Supplementary file-S3 There are only the names of the GPCRs and not the sequences. Is it normal ?

Supplementary file-S5 I am not familiar with the words network evidence et mutagenesis evidence. Be careful with the font style of network and mutagenesis evidence, there are different.

(d) Transmembrane helix detection : Why do you only check transmembrane domains in certain cases?

(f) Multiple Sequence : I didn’t find the supplementary Figure-S2

Results

The legend of Figure 2 can be longer. I don't understand why some items are surrounded by a big dotted circle.

(b) (i) Global analysis : Class A Rhodopsin with a capital letter because there are capital letters on Frizzled, Glutamate etc.

(b) (ii) Odorant : Why do you think there are 105 GPCRs in humans and only 6 in corals ? This can be further explained in the discussion. Perhaps you can talk about the GPCRs of taste and odorant in discussion.

(b) (iii) How can you explain that 2 GPCRs have more than 7 transmembrane domains ?

I didn’t found the Figures 3 and 4 in manuscript.

(b) (iv) - « Thus, we can conclude that this P. damicornis sequence indeed represents a functional opsin protein. » . I am embarrassed by the affirmative "we can conclude" because no functional test was carried out.

- « even distinguish colors » could be explained more.

(d) (i) TLRs in P. damicornis : Same remark with “After demonstrating that there are functional GPCRs” because in my opinion, functional proof is when a functional test is performed, even if bioinformatic evidence is advanced.

(d) (iii) PROSITE analysis : Figure 3A or Figure 7 ?

Discussion

Be careful, some italics are missing in the name of the species.

The discussion is shorter than the other parts. Perhaps you can discuss about the taste and odorant GPCRs. It is important to talk about the fact that the discovery of homologous receptors is done mainly through bioinformatics approaches, but it seems important to say that another approach, such as pharmacological, is necessary to confirm the hypotheses.

I understand that your method is to find homologs in non-model organisms using the genomes of model organisms (all vertebrates except Drosophila) but I have a few questions. Even if the genome of other non-model organisms is poorly annotated, it could be interesting to use them to refine research . I was wondering if there are TLRs in other species than vertebrates? Because by using only vertebrate TLRs, are we not missing homologous sequences in corals?

Reviewer #2: In this manuscript, the authors presented a pipeline to identify homologs of human membrane receptors in the coral P. damicornis. The authors argue that using this pipeline would help to find the right homologs and help to transfer knowledge accumulated from human-related research to the non-model organisms that were not well studied.

While I agree that the methods introduced here could help in homolog finding, I suggest that the authors re-consider some of their conclusions, such as “P. damiconis can smell, taste, and see light”. These kinds of conclusions are too subjective. Each of these “senses” is supported by a complex signaling cascade, finding a potential homolog of the related receptors is still far far away from such a conclusion.

Even when checking only on the homolog, the authors need to be careful with the words they are using. A lot of the context is referring to the function of a receptor, however, the real evidence the authors present is only the structure predictions. Although combing active site/region alignment with structural docking could provide some supports to the functional predictions, we usually take it only as a reference when working on structural biology. The real structure-function interpretation comes only from (at least near) physiological conditions. So I suggest that the authors do not overstate the results and put them into an “estimation” concept.

Other than the problems with overstating, the manuscript is not well organized. Most of the figures are mis-cited in the main text. In fact, only Figure 1 and 2 are cited correctly. And there is result appearing only in Discussions. Some of the writings are not scientific, for example, the second last paragraph in the Discussions. All these problems made the manuscript very hard to read.

6. PLOS authors have the option to publish the peer review history of their article (what does this mean?). If published, this will include your full peer review and any attached files.

Reviewer #1: No

Reviewer #2: No

---

## [Author Response · Author response to Decision Letter 0]

17 Jun 2022

We would like to thank the reviewers for their detailed and thoughtful feedback. We hope that we have enhanced readability (item 4 in comments) and adjusted our conclusions based on the data obtained more adequately (item 2 in comments). A point by point response to the concerns raised in item 5 in comments is provided below. All reviewer comments are shown in black font. All responses are shown in blue font. 

Reviewers' comments:

Reviewer's Responses to Questions

Comments to the Author

1. Is the manuscript technically sound, and do the data support the conclusions?

Reviewer #1: Yes

Reviewer #2: Partly

2. Has the statistical analysis been performed appropriately and rigorously? 

Reviewer #1: Yes

Reviewer #2: N/A

3. Have the authors made all data underlying the findings in their manuscript fully available?

Reviewer #1: Yes

Reviewer #2: Yes

4. Is the manuscript presented in an intelligible fashion and written in standard English?

Reviewer #1: Yes

Reviewer #2: No

5. Review Comments to the Author

Reviewer #1: The article « Transfer of knowledge from model organisms to evolutionarily distant non-model organisms : the coral Pocillopora damicornis membrane signaling receptome » discusses the difficulty in finding, with the genomic resources of model organisms, homologs in model organisms unconventional. The article proposes a bioinformatic approach, clearly detailled, in the form of a manual, which can be used in corals as in other non-model organisms, in order to find receptor homologs. This document opens with a long and rich introduction, written with clarity, thus facilitating comprehension by a non-bioinformatician. 

Comment #1: However, the legends of the figures could be more explicit in order to improve the reading and the comprehension of the document. 

Response #1: 

All figure legends except those for figures 6+7 have been expanded.

Comment #2: In my opinion, the discussion is short compared to the introduction and could be enriched. Despite these remarks, the article deserves to be published with minor corrections. Here are my comments :

Response #2: The discussion section has been largely rewritten. We have removed the recap of the findings (see comment from reviewer #2 to avoid the impression that results have been described in the discussion). This deleted approximately 1 page. Instead, we have now expanded drastically the biological interpretation of significance of the results, and have added statements placing our results into the context of other studies for which we have inserted numerous additional references. We have added 2 pages of new discussion, especially regarding the role of TLR in communication with microbes, the relatively small number of olfactory homologs in P. damicornis, especially as compared to sea urchins, the relationship between chemosensory capabilities and specialized tissues and organs, and the sensing of light using opsin proteins.

Introduction

Comment #3: - Add capital letters to : « Hidden Markov Models » instead « hidden Markov models »

Answer #3: correction has been made throughout the document

Material and methods

Comment #4: - Supplementary file-S2 I'm not sure I understand what the old and new membrane mean

Answer #4: Sorry for the confusion caused. The results section has been updated with specific references to the three column titles and a tab has been inserted in the excel file explaining the three datasets of membrane receptors compared. In short, we obtained 374 unique P. damicornis sequences from the 978 human plasma membrane receptome (older list, based on (Ben-Shlomo et al., 2003)), 329 from the 1352 human membrane proteome (newer list, based on (Almén et al., 2009)) and 151 from the 825 human GPCRs (based on (https://www.uniprot.org/docs/7tmrlist.txt)). These lists and their respective overlap are provided in supplementary file-S2, labeled “old membrane”, “new membrane” and “GPCR”, respectively. Furthermore, we added a short legend to the front tab:

for definitions of "old membrane", "new membrane" and "GPCR" see tab "description"

in short:

old membrane = 2003 list

new membrane = 2009 list

GPCR = uniprot list

For each P. damicornis id (column A), you can see if this sequence was found to be homologous to a protein in the old (green), new (yellow) or GPCR (red) list

An empty field indicates that it was not found in the respective list.

Comment #5:

- Supplementary file-S3 There are only the names of the GPCRs and not the sequences. Is it normal ?

Response #5: The sequences have been added.

Comment #6:

Supplementary file-S5 I am not familiar with the words network evidence et mutagenesis evidence. Be careful with the font style of network and mutagenesis evidence, there are different.

Response #6: Sorry for the confusion causes. "Network evidence" and "mutagenesis evidence" refer to the analyses we performed in Figures 5 and 6 respectively of the main text. We've added some descriptive text to Supplementary Table S5 to explain these columns, namely:

1 Network evidence refers to proteins which were identified as potential GPCR candidates by an analysis of PPI in homologous P. damicornis and M. capitata networks (Main Text Figure 5)

2 Mutagenesis evidence refers to proteins which were identified as potential GPCR candidates by an analysis of in silico mutation and predicted interaction score (Main Text Figure 6)

We've left the cells as saying "yes" or "no" since we don't have numerical cutoffs, it is largely based on a holistic assessment of the same alpha/beta/gamma subunit interactions being predicted in both species (Figure 5) or in a significant drop in mutated vs. unmutated sequence PPI prediction (Figure 6). 

Comment #7:

(d) Transmembrane helix detection : Why do you only check transmembrane domains in certain cases?

Response #7: Transmembrane helices were predicted in all cases but were only discussed in some cases where the prediction yielded additional conclusions that could not be reached with the alignment alone. The text in the methods description has been updated accordingly.

Comment #8:

(f) Multiple Sequence : I didn’t find the supplementary Figure-S2

Response #8. This is a typo that should read supplementary Figure-S1. The correction has been made. 

Results

Comment #9: 

The legend of Figure 2 can be longer. I don't understand why some items are surrounded by a big dotted circle.

Response#9: The figure legend has been expanded (and other legends as well, see comment #1). 

Comment #10:

(b) (i) Global analysis : Class A Rhodopsin with a capital letter because there are capital letters on Frizzled, Glutamate etc.

Answer #10: The correction has been made.

Comment #11:

(b) (ii) Odorant : Why do you think there are 105 GPCRs in humans and only 6 in corals ? This can be further explained in the discussion. Perhaps you can talk about the GPCRs of taste and odorant in discussion.

Answer #11: This is a very good question. We have discussed it in more than 1 page of added discussion and insertion of numerous references. We have also checked for discrepancies in the number of olfactory receptors in the GPCR list from uniprot and a published list of over 400 olfactory receptors (see paper by Trimmer et al from 2019). This paper includes many human variants making the list larger. The overlap between the uniprot id’s cited by Trimmer et al and the human subset of the uniprot list is 208 and the results section has been updated accordingly.

The new olfactory receptor section in discussion reads:

“Invertebrates possess remarkable chemosensory capabilities to explore and detect minute biochemical compounds. Despite the low number of odorant receptors found in P. damicornis, corals have demonstrated high sensitivity to chemical cues (e.g. (Tebben et al. 2015)Tebben et al 2015) and we postulate that this low number can be explained by the absence of organs dedicated to smell compared to higher organisms. Cnidarian polyp tentacles serve for prey capture and transport (to oral cavity), aggression and defense, and the cells they are made of (cnidocytes) also have various specialized functions (Fautin 2009) (e.g. Fautin, 2009) however, they constitute, with the oral cavity and the epidermis, the most specialized cnidarian sensing organ.

Generally, odors are used by organisms for survival purposes, such as homing, finding food, distinguishing between the same species and predators, predator avoidance and defense, and reproduction. In humans, the nose is a specialized organ to detect volatile compounds, and so we usually associate the ability to smell only with land animals. However, marine organisms can also smell. For example, fish have nostrils and pump water through them, and the compounds detected are decoded by their olfactory bulb. Fish can detect very low concentrations of compounds and use it for example to detect the direction of their home reef (Dixson et al., 2008). Anemones have long been known for their production of the pyridinium compound, amphikuemin, that attracts its symbiotic fish which swims towards it (Murata et al., 1986). In humans the detection and response to olfactory cues is so important that a remarkable gene expansion akin to that of antibody diversity has taken place giving rise to hundreds of olfactory receptors, which are highly diverse in different individuals underlying the differences between preferences and detection abilities in human populations (Trimmer et al., 2019). Although corals emit large numbers of volatile compounds (Lawson et al., 2020), for marine organisms, it is less important if compounds are volatile in air, as they would be dissolved in water, and so olfactory reception is better referred to as chemoreception in general. This can be well demonstrated with the sea urchin (Raible et al, 2006). For example, genomic clustering and single-exon gene structures suggest rapid gene duplication creating a new class of GPCRs, the “group B surreal-GPCR’s in the sea urchin that are differentially expressed in pedicellariae and tube feet, suggesting that these organs take up an analogous role to the human nose in sensing chemical stimuli (Raible et al., 2006). A similar rapid gene duplication gave rise to the diversity of olfactory receptors in humans, suggesting a similarity in the benefit of expansion of the organism’s capabilities to detect a larger diversity of ligands. These gene duplication events are highly organism specific, for example the holothurians are predicted to contain 57 olfactory receptors (Marquet et al. 2020). The next step after identification of the receptor repertoire is of course the identification and matching of corresponding ligands. Most progress to date with respect to ligands has been made with neuropeptides (Takahashi, 2020). Interestingly, there are larger numbers of putative taste receptors as compared to odorant receptors in P. damicornis (15 coral for 28 human taste receptors), which might indicate that sensing chemicals as part of the food intake machinery is an important chemosensory part of P. damicornis. “

Comment #12:

(b) (iii) How can you explain that 2 GPCRs have more than 7 transmembrane domains ?

Response #12:

Sorry for the misleading wording. These 2 GPCR’s are characterized by an additional large domain of length 500 amino acids before the 7 transmembrane domain. The text has been edited as follows:

“02512 and 09436 have a large N-terminal domain of approximately 500 amino acids preceding the more than 7 predicted transmembrane helical GPCR motif. This architecture is similar to what is found in human metabotropic glutamate receptors, where the soluble glutamate binding domain precedes the 7 transmembrane helical motif, but the added domains in the above two P. damicornis putative GPCR sequences show no similarity with the human glutamate binding domain.”

Comment #13:

I didn’t found the Figures 3 and 4 in manuscript.

Response #13: Sorry for the confusion. Figure numbers have been corrected and additional references to figures have been added throughout the results section.

Comment #14:

I don’t understand Figures 3 

Response #14: This is because we incorrectly referenced the figure numbers in the text and didn’t explicitly cite Figure 3 in the text that describes it. We now corrected this (see comment #13). We also expanded the explanation of Figure 3 and its figure legend.

Comment #15:

(b) (iv) - « Thus, we can conclude that this P. damicornis sequence indeed represents a functional opsin protein. » . I am embarrassed by the affirmative "we can conclude" because no functional test was carried out.

Response #15: We agree that this statement is too affirmative. We have rephrased it to say: “This finding supports the hypothesis”.

Comment #16

- « even distinguish colors » could be explained more.

Response #16: We have added a discussion of this statement to the discussion section, which reads as follows:

“ Much simpler than differentiating the diversity of chemical compounds is sensing the color and intensity of light, a task that can be accomplished by humans with 11 members of the opsin subfamily of class A GPCR. Again, P. damicornis shows a reduced repertoire: based on our detailed sequence and structural analysis of opsin candidates we predict 4 GPCR that may be able to bind retinal and gain light sensitivity in P. damicornis. This is consistent with the hypothesis that P. damicornis can likely sense and respond to light and even distinguish colors. Previous research showed that coral larvae preferentially settle on red as opposed to white surfaces (Foster & Gilmour, 2016). Consistent with this behavioral finding, is the inclusion of the red opsin protein (OPSR) in our list. On the other hand, there is also evidence for the ability to sense blue light - the action spectra of polyp retraction in the corals E. fastigiata and M. cavernosa have absorbance maxima of 482 and 478 nm, with half width at half height of 115 and 105 nm, respectively (Gorbunov and Falkowski, 2001). These narrow band widths support the conclusion of a single opsin involved in light sensing for polyp retraction in each coral, consistent with the small number of opsin candidates obtained for P. damicornis. The counter-ion of vertebrate opsin E is replaced by Y in invertebrate opsins in general (Kojima et al., 2017), also observed for P. damicornis, with the exception of S in 19775. The 19775 protein maps to both medium wave and long wave sensitive human opsins, further suggesting a change in absorbance maxima. Thus, despite the absence of eyes, which are only needed when spatial vision is needed, detection of light is a fundamental ability shared with many other organisms, such as sea urchins (Raible et al., 2006). Even when such organs are missing, the expression of specific opsin genes is not ubiquitous, providing a pathway towards encoding spatial information. “

Comment #17:

(d) (i) TLRs in P. damicornis : Same remark with “After demonstrating that there are functional GPCRs” because in my opinion, functional proof is when a functional test is performed, even if bioinformatic evidence is advanced.

Response #17. Agreed. Change has been made. It now reads “After our investigation of putative GPCRs”

Comment #18:

(d) (iii) PROSITE analysis : Figure 3A or Figure 7 ?

Comment #18: Yes, correction has been made.

Discussion

Comment #19

Be careful, some italics are missing in the name of the species.

Response #19: The correction has been made.

Comment #20

The discussion is shorter than the other parts. Perhaps you can discuss about the taste and odorant GPCRs. 

Response #20. A discussion on taste and odorant receptors has been added, also see response to comment #11.

Comment #21:

It is important to talk about the fact that the discovery of homologous receptors is done mainly through bioinformatics approaches, but it seems important to say that another approach, such as pharmacological, is necessary to confirm the hypotheses.

Response #21: We agree, and we have added the following section to the discussion:

“Although we restricted our detailed structure-function analysis to the GPCR subfamily of opsins in vision, G proteins, and TLRs, we also presented some initial analysis of possible odorant, taste and other receptors in corals. It is important to note, however, that the discovery of homologous receptors is done mainly through bioinformatics approaches, and thus can only be viewed as hypothesis generating work. Experimental approaches, such as pharmacological, cell biological and/or biochemical, are necessary to confirm these hypotheses. Thus, while further functional and especially experimental tests are needed to verify these predictions, the approach presented identifies potential new targets for studying coral biology and possibly draw connections between the general functions of the human proteins that the coral proteins mapped to and their potential role in coral sensing. “

Comment #22:

I understand that your method is to find homologs in non-model organisms using the genomes of model organisms (all vertebrates except Drosophila) but I have a few questions. Even if the genome of other non-model organisms is poorly annotated, it could be interesting to use them to refine research . I was wondering if there are TLRs in other species than vertebrates? Because by using only vertebrate TLRs, are we not missing homologous sequences in corals?

Response #22: Yeast is not known to express TLRs. However, we added analysis of the single C. elegans receptor known for innate immunity regulation (TOL-1), uniprot id Q9BIW9. All figures and text are updated accordingly. We decided to not include an evolutionary analysis of TLR’s in other invertebrates, but we included a discussion of studies involving TLR’s in Nematostella, and Acropora in the updated discussion. 

Reviewer #2: In this manuscript, the authors presented a pipeline to identify homologs of human membrane receptors in the coral P. damicornis. The authors argue that using this pipeline would help to find the right homologs and help to transfer knowledge accumulated from human-related research to the non-model organisms that were not well studied.

Comment #23:

While I agree that the methods introduced here could help in homolog finding, I suggest that the authors re-consider some of their conclusions, such as “P. damiconis can smell, taste, and see light”. These kinds of conclusions are too subjective. Each of these “senses” is supported by a complex signaling cascade, finding a potential homolog of the related receptors is still far far away from such a conclusion.

Response #23: We completely agree, and have replaced such statements with more cautious ones throughout the manuscript. For example “can smell, taste and see light” was replaced by “corals have sensory receptors analogous with smell, taste and light perception”, “conclude” by “supports the hypothesis that” etc.

Comment #24:

Even when checking only on the homolog, the authors need to be careful with the words they are using. A lot of the context is referring to the function of a receptor, however, the real evidence the authors present is only the structure predictions. Although combing active site/region alignment with structural docking could provide some supports to the functional predictions, we usually take it only as a reference when working on structural biology. The real structure-function interpretation comes only from (at least near) physiological conditions. So I suggest that the authors do not overstate the results and put them into an “estimation” concept.

Response #24: We agree and have now explicitly stated the use of structure to infer function and described for each use of the word “function” the context of how it should be interpreted.

Comment #25

Other than the problems with overstating, the manuscript is not well organized. Most of the figures are mis-cited in the main text. In fact, only Figure 1 and 2 are cited correctly. 

Response #25:

We apologize for the mess created with the figure numbering. These have all been corrected.

Response #26:

And there is result appearing only in Discussions. 

Comment #26: Reference to results has been deleted from the discussion (most of the original first page of the discussion has thus been deleted).

Comment #27:

Some of the writings are not scientific, for example, the second last paragraph in the Discussions. 

Response #27: The paragraph has been rewritten. 

Comment #28

All these problems made the manuscript very hard to read.

Response #28: We have substantially edited the entire manuscript and believe that it has become easier to read. We would like to thank both of the reviewers for their constructive comments which have allowed us to generate a much-improved manuscript.

---

## [Editor Report · Decision Letter 1]

22 Jun 2022

Transfer of Knowledge from Model Organisms to Evolutionarily Distant Non-Model Organisms: The Coral Pocillopora damicornis Membrane Signaling Receptome

PONE-D-21-34522R1

Dear Dr. Klein-Seetharaman,

We’re pleased to inform you that your manuscript has been judged scientifically suitable for publication and will be formally accepted for publication once it meets all outstanding technical requirements.

Kind regards,

Karl-Wilhelm Koch, Ph.D.

Academic Editor

PLOS ONE

---

## [Editor Report · Acceptance letter]

15 Sep 2022

PONE-D-21-34522R1 

Transfer of Knowledge from Model Organisms to Evolutionarily Distant Non-Model Organisms: The Coral *Pocillopora damicornis* Membrane Signaling Receptome 

Dear Dr. Klein-Seetharaman:

I'm pleased to inform you that your manuscript has been deemed suitable for publication in PLOS ONE. Congratulations! Your manuscript is now with our production department. 

Kind regards, 

on behalf of

Dr. Karl-Wilhelm Koch 

Academic Editor

PLOS ONE